# The Potential of Termite Mound Spreading for Soil Fertility Management under Low Input Subsistence Agriculture

Amsalu Tilahun [1,*], Wim Cornelis [2], Steven Sleutel [2], Abebe Nigussie [1], Bayu Dume [3] and Eric Van Ranst [4]

1   Department of Natural Resources Management, Jimma University, Jimma P.O. Box 307, Ethiopia; abenigussie@gmail.com
2   Department of Environment, Ghent University, Coupure Links 653, 9000 Ghent, Belgium; wim.cornelis@ugent.be (W.C.); steven.sleutel@ugent.be (S.S.)
3   Department of Agro-Environmental Chemistry and Plant Nutrition, Faculty of Agrobiology, Food, and Natural Resources, Czech University of Life Sciences, Kamycka 129, 16500 Prague, Czech Republic; dumebayu@gmail.com
4   Department of Geology (WE13), Ghent University, Campus Sterre S8, Krijgslaan 281, 9000 Ghent, Belgium; eric.vanranst@ugent.be
*   Correspondence: amsetilahun@gmail.com

**Abstract:** Termites can play a localized prominent role in soil nutrient availability and cycling because mound materials are often enriched in nutrients relative to surrounding soil. Mound materials may thus prove to be useful amendments, though evidently mound spatial arrangement needs to be considered as well. Furthermore, it is not known if gradients of soil properties exist from termite mound to interspace sites. Studying both aspects would be required to decide whether spreading of mounds or spatially differentiated management of surrounding crop to accommodate soil fertility gradients would be valid nutrient-management strategies. Mound abundance and mass were estimated at 9 and 4 mounds ha$^{-1}$, representing 38.9 and 6.3 t ha$^{-1}$ on Nitisols and Vertisols, respectively. Soil physical and chemical properties were measured on samples collected from internal and external parts of mounds and adjacent soils at 0.5, 1 and 10 m away from mounds. In general, termite mounds were enriched in plant nutrients and SOC on Vertisols but not on Nitisols. Termite mounds constituted only 0.3 to 1.3% of the 0–15 cm SOM stock on a per ha basis but nevertheless the immediate vicinity of termite mounds was a relative fertile hotspot. Hence, under the studied condition, we suggest spatial arrangement of crop around termite mounds according to soil fertility gradient and spatially differentiated nutrient management strategies. Our result suggests recommendation of termite mound spreading for soil nutrient amendment has to consider plant nutrient stock in termite mounds on per ha basis besides their nutrient enrichment. Interesting topics for future investigation would be growth experiment for different crops with mound materials treatment. It would also be interesting to study the effect mound building termite on soil properties under different soil conditions, slope class and land use.

**Keywords:** termite mounds; Vertisols; Nitisols; interspace site; mounds spreading; aggregate stability

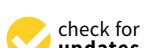



## 1. Introduction

Current farming mainly uses mineral fertilizer to boost agronomic yield [1]. Mineral fertilizers substantially increased crop yield especially after green revolution. However, the average application rate of mineral fertilizer is tenfold lower in sub-Saharan Africa than in other continents [2]. In Ethiopia, over 40% of the soils are acidic and two thirds of the soils are characterized by phosphorous (P) deficiency [3]. Common reasons for low fertilizer application are: (i) the majority of African subsistence farmers are unable to afford mineral fertilizer because of its high price [4,5], (ii) farmers' limited access to inputs because of poor infrastructure [6], (iii) and poor response of crops to mineral fertilizer because of moisture stress [5]. Consequently, low crop productivity is a general problem facing most

farming systems in Sub- Saharan Africa [7]. Too high fertilizers cost obstructed a successful 'green revolution' among many smallholder farmers, who are nevertheless responsible for the majority of food production [8,9]. Furthermore, there is general consensus that the nonrenewable rock phosphate is depleting and unable to meet future demand for P fertilizer [10,11]. Consequently, research efforts have directed to explore more choices that are locally available and economically sound. For example, the Alliance for a Green Revolution in Africa (AGRA) has forwarded Integrated Soil Fertility Management (ISFM) that advocates the maximum use of locally available resources. Recently, research efforts have directed to explore more choices that are locally available and economically sound. Biological nitrogen fixation for improving soil fertility and food security [12], biochar for soil quality and crop yield [13–15], conservation agriculture [16], input of native plant biomass [17], and termite mound materials as amendment for soil amelioration [18,19].

Termites and earthworms can be considered as an important ecosystem engineers in soil, because of their far reaching and lasting effects on soil physical and chemical properties [20]. Through their mound-building activities, termites inevitably cause movement of substantial amounts of soil particles within and above the soil surface [18,21–23]. Their role in soil nutrient availability and cycling at various spatial and temporal scales is significant [24]. Comparative studies of termite mounds and the adjacent termite-unmodified control soils reported usually higher concentration of organic matter (OM) and mineral nutrients in termite mounds than in reference soil [25–28]. This led to suggest mound materials as a soil fertility amendment in smallholder farming [18]. Accurate quantification of mounds soil mass and estimation of the amount of nutrients stored per area are however much needed to evaluate the actual potential of mound spreading and sustainability of such practice.

Erosion and human action eventually redistribute mound material to the surrounding soils, causing changes in soil microstructure and fertility non-uniformly manifested over spatial gradients [29]. Despite this fact, most of the studies that link termite mound effects on soil compare pure mound materials with termite-unmodified reference soils, leading to biased assessment of the potential of these amendments for soil modification. Detailed empirical studies on gradients of soil physical and chemical properties that exist from termite mound to interspaces sites have not been conducted. Our objective was thus to accurately and comprehensively quantify nutrients stored in termite mounds and spatial variability of soil around termite mounds. Such study should help deciding whether to spread mounds, to which spatial extent mounds spreading should be completed, and to plant suitable crops around termite mounds and guide nutrient-management strategies on the basis of niche fertility. Secondly it is at present speculative to predict how soil type interacts with the effect of mound material spreading on soil fertility. For instance, Nitisols are deep, well-drained and is highly clay dominated soils by kaolinite and (meta) halloysite having high iron and aluminum oxides. The high iron and aluminum oxides lead to widespread P deficiency due to strong fixation. If mound materials are enriched in partially available P sources spreading out this material may critically raise agricultural productivity. On the other hand, Vertisols are characterized by heavy clay, swelling when wet and cracking when dry but chemically fertile soils naturally. But poor drainage and difficult workability limit nutrient availability. Tillage and seedbed preparation are only possible within a narrow soil-moisture range. These soils remain uncultivated during part of the rainy season, because they waterlog, and many highland crops such as teff, barley, durum wheat, chickpea, lentil, noug and vetch are grown on residual moisture at the end of the rains. Spreading out of the commonly OM-enriched mound material may critically lift locally soil physical quality. Both Nitisols and Vertisols are common in Jimma, Ethiopia covering 23 and 18% respectively. Although mound-building termites are widespread, they are often considered as pests. Consequently, research has even been concentrated on the pest management aspect. For these reason that we undertook a comprehensive study on physical and chemical properties of the mounds in relation to the surrounding soils and accurately quantified the nutrient stored in termite mounds to evaluate the potential

of termite mounds as an alternative means of soil management in Jimma area, South west Ethiopia.

## 2. Materials and Methods

### 2.1. Site Description

The study was conducted in Jimma Zone in the South west of Ethiopia far away about 360 km from Addis Ababa. Soils and termite mound materials were sampled on Nitisols at Limmu kosa ($7°50'$–$8°36'$ N and $36°44'$–$37°29'$ E, 1657 m.a.s.l) and Vertisols at Omo Nadda ($7°17'$–$7°49'$ N and $37°00'$–$37°28'$ E, 1593 m.a.s.l) (Figure 1). The Climate is humid, subtropical with a peak rainfall occurring between mid-June to mid-September (long rainy season) and within a smaller (short rainy season) from February to May, with mean annual rainfall of 2000 and 1700 mm at Limmu kosa and Omo Nadda, respectively. Temperature is fairly constant throughout the year, with the mean minimum, maximum and average temperatures being 11, 25 and 17 °C, respectively. The major reference soil groups at Omo Nadda are Acrisols, Vertisols and Planosols. Vertisols, Acrisols and Nitisols are the major soil types of Limmu kosa district. In both study sites, the landscape is characterized by dissected plateaus, mountains, hills, plains and valleys. The farming system of the study sites is characterized by mixed crop-livestock subsistence farming. Constraints to agricultural practices are low soil fertility, land degradation, lack of access to modern technology and marketing. Termite mounds are widespread in both study areas and farmers largely complain the damage of termites. Morphologically, mounds of the study areas are characterized as low and had dome to more flattened shapes on top with no external openings. Based on species identification done by [30] using termite mounds external morphology, termites in both study areas (Omo Nadda and Limmu kosa) are grouped into genus *Macrotermes* and *M. herusspecies*.

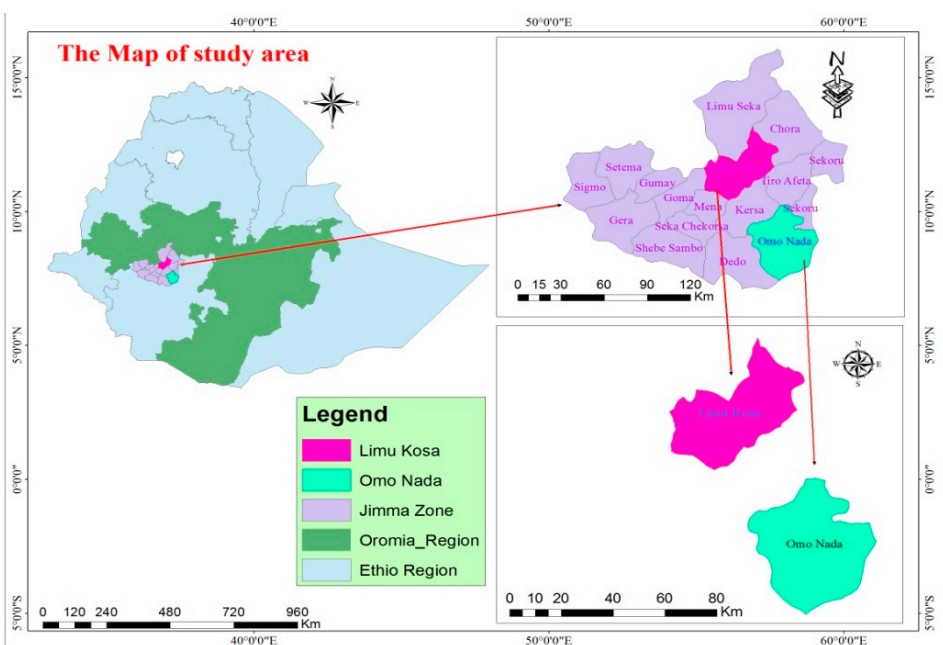

**Figure 1.** Map of study area.

### 2.2. Soil Sampling

There is no standard sampling protocol from termite mounds and their corresponding adjacent termite-unmodified reference soils. For example, soil samples collected at distance of 1 m [31], at 1.5 m [27], at 5 m [32] and at 10 m [33] were considered as reference soil. For this study, soil samples were separately collected from internal and external parts of termite mounds and from adjacent soils at 0.5, 1 and 10 m distances. Soil samples collected at 10 m away from the mounds were considered as termite-unmodified soils, here defined

as reference soil without significant termite activity. Composite 2 kg samples were collected in four directions from termite mounds. Undisturbed samples were collected by 5 cm diameter and 5.1 cm height Kopecky rings. On both soil types, three termite mounds were randomly chosen as replicates. Soil samples were collected from active termite mounds.

### 2.3. Soil Chemical Properties

The soil samples were air-dried, mixed well and passed through a 2 mm sieve for analysis. Particle size analyses were performed using the hydrometer method. Bulk density was determined after drying the core samples in an oven at 105 °C for 24 hrs. Soil pH ($H_2O$) was measured using pH-meter in a 1:2.5 soil: water mixture. Soil organic carbon content was measured by the Walkley and Black wet digestion procedure [34] and total nitrogen was determined by the Kjeldhal method [35]. Readily plant-available phosphorous, i.e P-intensity was determined by the Bray-1 extraction method. P bound to Fe and Al (hydr-)oxides, i.e., P-capacity, was determined by acid $NH_4$-oxalate extraction, along with the ratio of P to extracted Al and Fe ($Fe_{ox}$ and $Al_{ox}$). Determination of total phosphorous ($P_{tot}$) was carried out according to [36]. Exchangeable cations (Ca, Mg, K and Na) were determined by extraction using 1 M ammonium acetate. Extracts were analyzed for their Fe, Al, Ca, Mg, K, Na and P contents by a 6300 radial iCAP ICP-OES (Thermo Scientific, Santa Clara, CA, USA). Cation exchange capacity was determined at pH 7 after displacement by using 1 N ammonium lactate method; washing with ethanol and final extraction of the exchanged $NH_4$ by 1 M KCl. KCl extracts were determined for their $NH_4^+$ content by means of continuous-flow auto-analyser (Skalar, Breda, The Netherlands)

### 2.4. Soil Physical Properties

The soil water retention curve (SWRC) was determined on undisturbed Kopecky ring soil cores, by measuring soil moisture content at soil matric potentials of −1, −3, −5, −7, −10, −33, −100 and −1500 kPa following the procedure described by [37]. Undisturbed soil samples were weighed and saturated gradually from below prior to placement on to the sand box (EijkelkampAgrisearch Equipment, Giesbeek, the Netherlands) in order to determine moisture content at matric potentials of −1, −3, −5, −7 and −10 kPa. The pressure chamber (soil moisture Equipment, Santa Barbara, CA, USA) was used to determine moisture content for lower pressures (−33, −100 and −1500 kPa). Plant available water (PAW) was calculated as the difference between field capacity (−10 kPa) and permanent wilting point (−1500 kPa).

Aggregate stability was measured on air-dried soil samples using the method of [38]. Slow wetting (SW) of air-dried soil samples were done by capillary force on a tension table at a matric potential of –0.3 kPa. Whereas in fast wetting (FW) air-dried soil samples were directly put to wet sieving in deionized water. 4 g of 1–2 mm diameter aggregates were placed on a sieve with mesh size of 0.25 mm (diameter) and sieved in a can containing distilled water for 3 min at constant, automatically controlled speed using a device commercially available as wet sieving apparatus by EijkelkampAgrisearch Equipment (EijkelkampAgrisearch, Giesbeek, the Netherlands). Unstable aggregates that passed through the sieve were collected in a cup and their weight was determined after oven drying. Soil that remains on the sieve after mechanical shaking was considered as stable aggregates. Dispersed aggregates that passed through the sieve of 0.25 mm with deionized water and in a solution of sodium metaphosphate were collected in a cup and weighed after oven drying. Analyses were replicated three times. Results were expressed in terms of the mean weight diameter (MWD) and percentage of Water Stable Aggregates (*WSA*) as:

$$MWD(mm) = \frac{\sum_{i=1}^{i=n} m_i d_i}{m_t} \tag{1}$$

where $m_i$ is mass of the stable aggregate fraction i; $d_i$ is mean diameter of fraction i; $m_t$ is total weight of the sample, and

$$WSA = \frac{Wds}{(Wds + Wdw)} \tag{2}$$

where *WSA* is the index of water stable aggregates, *Wds* is the weight of aggregates dispersed in dispersing solution (g), and *Wdw* is the weight of aggregate dispersed in distilled water (g).

### 2.5. Estimation of Mounds Abundance and Potential of Mound Materials

The number of termite mounds was counted in areas of 10 and 7 ha in the Limmu kosa and Omo Nadda districts, respectively. The areas were stratified into four blocks based on the pattern of mounds arrangement. In each of the stratified blocks, three quadrants of 50 m × 50 m were laid out randomly. Thus, a total of 12 quadrants for one study site were counted for number of mounds to estimate mounds abundance per hectare. All mounds were described morphologically in terms of height, circumference, estimated basal area, volume and mass. The radius was calculated from the measured circumferences. The volume was determined assuming that the mounds approximated a shape of truncated cone with the equation

$$V = \pi H/3 \left[ r1^2 + r1r2 + r2^2 \right] \tag{3}$$

where *H* (m) is the thickness, *r*1 and *r*2 (m) are radii of the ends and *V* ($m^3$ $mound^{-1}$) is volume of specific mound. Finally, to quantify mass of mounds a regression equation developed by [18] was used accounting for numerous internal tunnels/voids in mounds. Mere calculation using volume and bulk density would overestimate mass of mounds by about 10% [18]. Nutrients available as alternative fertilizer were calculated by multiplying mound mass per hectare and nutrient contents.

### 2.6. Data Analysis

One-way ANOVA was performed with SPSS 21.0 (IBM, Armonk, NY, USA) to detect significant differences between the means of mound material and adjacent soils for all measured physical and chemical parameters. Tukey-Kramer HSD post-hoc test was used to identify significantly different means at $\alpha = 5\%$. Normality was tested with the Sharipo-Wilk test.

## 3. Results

### 3.1. Termite Mound Density and Characteristics

The termite mounds on both soil types had identical morphology, typically consisting of a cone-shaped base and frustum of cone at the top, attaining average height of 0.57 and 0.71 m, basal circumferences of 3.27 m (radii 0.97 m) and of 12.55 m (radii 2.00 m), volume above ground 5.69 and 39.23 $m^3$ $ha^{-1}$ and basal area of 14 and 119.50 $m^2$ $ha^{-1}$ on Nitisols and Vertisols, respectively (Table 1). The number of mounds in a quadrant of 50 m by 50 m ranged from 0 to 5 and 0 to 3, equivalent to a density of 9 and 4.33 mounds $ha^{-1}$, occupying 1.15 and 0.15% of land area on Nitisols and Vertisols, respectively. Calculating the total mass of nests from these densities and the mass of mounds using equation of [18], this equates to 6.3 t $ha^{-1}$ and 38.9 t $ha^{-1}$ mound materials on Vertisols and Nitisols, respectively. No stored fragments of leaves, grasses, twigs and finely divided unidentifiable organic materials in mounds. In contrast to common reports, termite mounds at both sites were relatively not cemented, not too hard to scratch and were easily augured. This was also evidenced by the growth of vegetation on the surface of the mounds that also partly contributed to loosening of the external parts of the termite mounds, along with weathering and erosion with time.

**Table 1.** Morphological description of termite mounds (mean + standard error of the mean (SEM)).

| Mound Characteristics | Omo Nadda, Vertisols (*n* = 22) | | | Limmu Kosa, Nitisols (*n* = 19) | | |
|---|---|---|---|---|---|---|
| | Minimum | Mean ± SEM | Maximum | Minimum | Mean ± SEM | Maximum |
| Height (m) | 0.3 | 0.7 ± 0.2 | 1.1 | 0.2 | 0.6 ± 0.2 | 1 |
| Basal area ($m^2$) | 0.8 | 3.2 ± 1.5 | 10.9 | 3.1 | 13.2 ± 4.9 | 31.9 |
| Basal area ($m^2$ $ha^{-1}$) | 3.6 | 14.0 ± 6.5 | 47.2 | 27.5 | 119.0 ± 44.8 | 286.7 |
| Volume ($m^3$) | 0.2 | 1.3 ± 0.7 | 4.5 | 0.4 | 4.4 ± 2.6 | 14.2 |
| Mass kg per mound | 139 | 1454 ± 8 | 5033 | 327 | 4320 ± 2900 | 14,134 |
| Mass per ha (t $ha^{-1}$) | 0.6 | 6.3 ± 3.3 | 21.8 | 2.9 | 38.9 ± 23.5 | 127.2 |

*3.2. Physical Soil Properties*

None of the analyzed physical properties differed significantly between the internal and external parts of termite mounds. On discussion section, we compared the average values of both with that of adjacent soils. On both soil types, clay content was significantly higher in termite mounds compared to adjacent soils (Table 2). But the enrichment was stronger on Nitisols with relative increases of 1.87, 2.13 and 1.92 times than on Vertisols with only 1.13, 1.19 and 1.16 times increase at 0.5, 1 and 10 m away from termite mounds, respectively. The bulk density ($\rho_b$) did not differ significantly. The soil aggregate stability indices discriminated between the termite mounds and adjacent soils (Table 3). Soil aggregate stability of termite mound materials was generally reduced compared to the control surrounding area, where in sampling distance was of no further influence.

**Table 2.** Physical properties of termite mounds and adjacent soil (means ± standard error of mean).

| Sites | Clay (%) | Silt (%) | Sand (%) | USDA Textural Class | $\rho_b$ ($Mgm^{-3}$) |
|---|---|---|---|---|---|
| | Omo Nadda, Vertisols | | | | |
| Mint | 60.7 ± 2.7 ba | 23.3 ± 2.7 a | 16.0 ± 0.0 a | Clay | 1.29 ± 0.0 a |
| Mext | 58 ± 2.3 ab | 23.3 ± 2.7 a | 18.7 ± 1.3 a | Clay | 1.21 ± 0.0 a |
| Adj. 0.5 m | 52.67 ± 0.7 ab | 28.0 ± 1.2 a | 19.3 ± 1.3 a | Clay | 1.14 ± 0.0 a |
| Adj. 1 m | 50.0 ± 1.2 a | 30.0 ± 1.2 a | 20.0 ± 0.0 a | Clay | 1.16 ± 0.1 a |
| Adj. 10 m | 51.1 ± 3.3 ab | 30.7 ± 0.7 a | 18.0 ± 3.2 a | Clay | 1.25 ± 0.0 a |
| *p*-value | * | NS | NS | | NS |
| | Limmu kosa, Nitisols | | | | |
| Mint | 80.0 ± 1.2 c | 15 ± 0.7 a | 5 ± 1.3 a | Clay | 1.10 ± 0.0 a |
| Mext | 73.3 ± 4.7 c | 17 ± 1.3 a | 9 ± 3.3 ab | Clay | 1.11 ± 0.1 a |
| Adj. 0.5 m | 41.0 ± 1.76 ab | 31 ± 1.3 c | 27 ± 0.7 bc | Clay | 1.02 ± 0.1 a |
| Adj. 1 m | 36.0 ± 1.2 a | 32 ± 1.2 c | 32 ± 2.0 cd | Clay L. | 1.08 ± 0.0 a |
| Adj. 10 m | 40.0 ± 1.2 ab | 31 ± 2.7 bc | 29 ± 3.7 c | Clay L. | 1.07 ± 0.0 a |
| *p*-value | * | * | * | | NS |

Means within a column followed by the different lowercase letters are significantly different at *p* = 0.05; Mint = internal part of the termite mound; Mext = the external part of the termite mound; Adj. 0.5 m, Adj. 1 m, Adj. 10 m represents adjacent soil at the distances of 0.5, 1 and 10 m from the termite mound respectively. * represents significant at *p* < 0.05; NS = not significant at *p* < 0.05.

There were no significant differences in field capacity (FC), permanent wilting point (PWP), plant available water (PAW) or porosity between sampled locations. Water retained at FC and PWP tended to be higher in termite mound material than of the adjacent soils, particularly on Vertisols (Table 4). Plant available water followed a similar trend on Vertisols, but not on Nitisols.

**Table 3.** Aggregate stability indices of termite mounds and adjacent soil (means ± standard error of mean).

| Sites | WSA-FW | WSA-SW | MWD-FW | MWD-SW |
|---|---|---|---|---|
| | Omo Nadda, Vertisols | | | |
| Mint | 71.5 ± 4.4 a | 89.1 ± 1.6 ab | 0.59 ± 0.1 a | 0.9 ± 0.0 a |
| Mext | 77.4 ± 1.5 ab | 83.2 ± 0.6 a | 0.75 ± 0.1 ab | 1.0 ± 0.0 a |
| Adj. 0.5 m | 83.2 ± 3.7 ab | 92.7 ± 2.1 b | 0.86 ± 0.1 b | 1. ± 0.1 a |
| Adj. 1 m | 88.1 ± 1.1 b | 94.7 ± 0.8 b | 0.98 ± 0.0 b | 1.0 ± 0.0 a |
| Adj. 10 m | 84.1 ± 5.1 ab | 92.0 ± 2.5 b | 0.91 ± 0.1 b | 0.9 ± 0.0 a |
| *p*-value | * | * | * | NS |
| | Limmu kosa, Nitisols | | | |
| Mint | 66.8 ± 5.6 a | 98.3 ± 0.4 a | 0.73 ± 0.1 a | 1.1 ± 0.0 a |
| Mext | 66.1 ± 2.5 a | 97.8 ± 0.4 a | 0.73 ± 0.1 a | 1.1 ± 0.0 a |
| Adj. 0.5 m | 96.6 ± 1.1 b | 99.4 ± 0.3 a | 1.05 ± 0.0 b | 1.1 ± 0.0 a |
| Adj. 1 m | 95.3 ± 0.8 b | 99.5 ± 0.3 a | 0.98 ± 0.1 b | 1.1 ± 0.0 a |
| Adj. 10 m | 96.0 ± 1.4 b | 99.5 ± 0.0 a | 1.00 ± 0.1 b | 1.0 ± 0.1 a |
| *p*-value | * | NS | * | NS |

Means within a column followed by the different lowercase letters are significantly different at $p = 0.05$; WSA-FW = percent of water stable aggregates after fast wetting, WSA-SW = percent of water stable aggregates after slow wetting; MWD-FW = mean weight diameter (mm) after fast wetting; MWD-SW = mean weight diameter (mm) after slow wetting. Mint = internal part of the termite mound; Mext = the external part of the termite mound; Adj. 0.5 m, Adj. 1 m, Adj. 10 m represents adjacent soil at the distances of 0.5, 1 and 10 m from the termite mound respectively. * represents significant at $p < 0.05$; NS = not significant at $p < 0.05$.

**Table 4.** Effects of termite mound on plant available water and porosity.

| Site | FC (*v/v*) | PWP (*v/v*) | PAW (*v/v*) | Porosity |
|---|---|---|---|---|
| | Omo Nadda, Vertisols | | | |
| Mint | 0.53 ± 0.02 a | 0.32 ± 0.00 a | 0.21 ± 0.03 a | 51.21 ± 0.71 a |
| Mext | 0.52 ± 0.03 a | 0.30 ± 0.01 a | 0.22 ± 0.02 a | 54.23 ± 1.30 a |
| Adj. 0.5 m | 0.47 ± 0.03 a | 0.29 ± 0.02 a | 0.17 ± 0.01 a | 57.01 ± 1.15 a |
| Adj. 1 m | 0.44 ± 0.01 a | 0.31 ± 0.02 a | 0.13 ± 0.02 a | 56.13 ± 1.85 a |
| Adj.10 m | 0.43 ± 0.05 a | 0.29 ± 0.06 a | 0.14 ± 0.00 a | 52.71 ± 1.48 a |
| *p*-value | NS | NS | NS | NS |
| | Limmu kosa, Nitisols | | | |
| Mint | 0.40 ± 0.02 a | 0.30 ± 0.01 a | 0.10 ± 0.01 a | 58.31 ± 0.12 a |
| Mext | 0.41 ± 0.02 a | 0.28 ± 0.01 a | 0.13 ± 0.01 a | 58.17 ± 2.66 a |
| Adj. 0.5 m | 0.40 ± 0.01 a | 0.27 ± 0.01 a | 0.13 ± 0.01 a | 61.41 ± 1.70 a |
| Adj. 1 m | 0.40 ± 0.02 a | 0.26 ± 0.01 a | 0.14 ± 0.01 a | 59.15 ± 0.47 a |
| Adj.10 m | 0.39 ± 0.01 a | 0.25 ± 0.00 a | 0.14 ± 0.01 a | 59.68 ± 0.63 a |
| *p*-value | NS | NS | NS | NS |

Means within a column followed by the different lowercase letters are significantly different at $p = 0.05$; FC = field capacity; PWP = permanent wilting point; PAW = plant available water; Values are means ± standard deviations of three replicates. Mint = internal part of the termite mound; Mext = the external part of the termite mound; Adj. 0.5 m, Adj. 1 m, Adj. 10 m represents adjacent soil at the distances of 0.5, 1 and 10 m from the termite mound respectively. * represents significant at $p < 0.05$; NS = not significant at $p < 0.05$.

In the Nitisols at matric potentials −1, −3 and −5 kPa water retention was relatively higher in adjacent soils than in termite mounds. Beyond −5 kPa, water retention was higher in the termite mounds (Figure 2). On Vertisol, at all measured matric potentials water retention was generally higher in termite mounds than the adjacent soils (Figure 2).

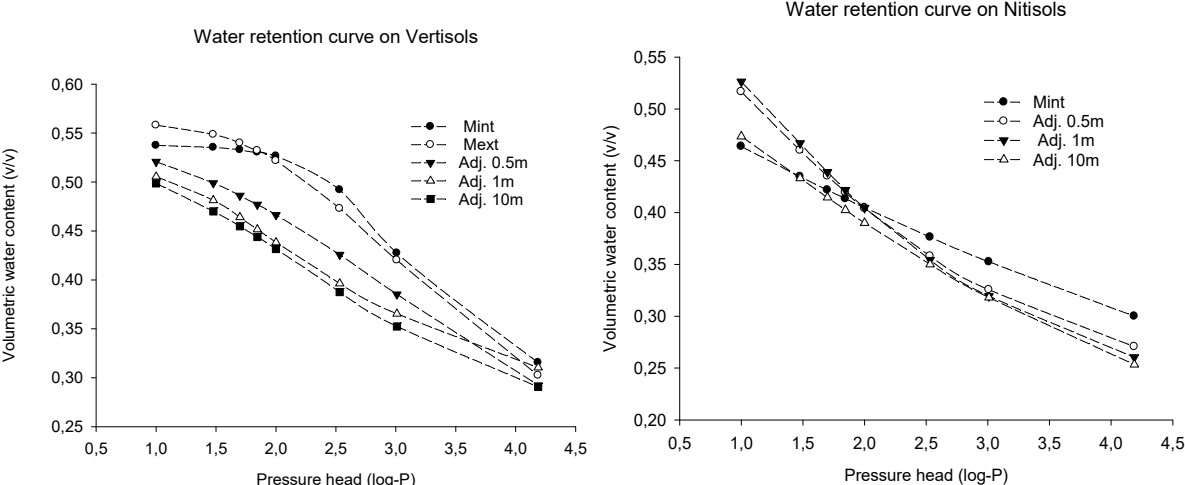

**Figure 2.** Soil water retention curves for termite mounds (internal and external parts) and adjacent soils at distances of 0.5, 1 and 10 m from termite mounds on Vertisols (**left**) and Nitisols (**right**) panel. Mint = internal termite mound; Mext = external termite mound; Adj. 0.5 m, Adj. 1 m, Adj. 10 m represents adjacent soil at distance of 0.5, 1 and 10 m from the termite mound respectively.

### 3.3. Soil Organic Carbon and Total Nitrogen and Plant-Available Nutrients

For both soil types, SOC content of termite mound material was lower ($p < 0.05$) than that of adjacent soil (Table 5). Soil samples collected at 0.5 and 1 m from the mounds had higher SOC content than the reference soil at 10 m but these differences were only significant at ($p < 0.05$). Hence, our results confirmed that on Vertisols areas within a radius of 1 m from mounds are SOC hot spots, with average increases of 79% and 19% compared to the SOC in termite mounds and reference soils, respectively. On Nitisols, the average increase was 100% and 20%, respectively. Total nitrogen (Nt) followed a similar trend as SOC content (Table 5). From the calculated mound soil masses and their SOC and Nt content, an average of 639 kg C ha$^{-1}$ and 43 kg N ha$^{-1}$ were contained in termite mounds on Nitisols and 102 kg C ha$^{-1}$ and 6 kg N ha$^{-1}$ on Vertisols. These correspond to 1.25% and 0.27% of the SOC stock in the upper 15 cm soil layer on Nitisols and Vertisols, respectively.

**Table 5.** Mean values of some chemical properties of mound materials and adjacent soils.

| Sites | pH (H$_2$O) | SOC (%) | Nt (%) | P$_{av}$ (mg kg$^{-1}$) |
|---|---|---|---|---|
| | | Omo Nadda, Vertisols | | |
| Mint | 7.1 ± 0.23 a | 1.68 ± 0.27 a | 0.12 ± 0.02 a | 6.53 ± 0.29 a |
| Mext | 7.1 ± 0.12 a | 1.53 ± 0.50 a | 0.08 ± 0.00 a | 6.76 ± 0.11 a |
| Adj.0.5 m | 6.8 ± 0.06 a | 2.88 ± 0.48 b | 0.13 ± 0.01 a | 11.33 ± 1.42 b |
| Adj. 1 m | 6.6 ± 0.09 a | 2.79 ± 0.012 b | 0.14 ± 0.01 a | 9.37 ± 0.27 ab |
| Adj. 10 m | 6.9 ± 0.10 a | 2.40 ± 0.42 ab | 0.10 ± 0.01 a | 5.34 ± 0.77 a |
| *p*-value | NS | * | NS | * |
| | | Limmu kosa, Nitisols | | |
| Mint | 5.6 ± 0.20 ab | 1.49 ± 0.04 a | 0.12 ± 0.01 a | 0.00 ± 0.00 a |
| Mext | 5.6 ± 0.12 ab | 1.79 ± 0.57 bc | 0.10 ± 0.01 a | 0.19 ± 0.19 a |
| Adj.0.5 m | 5.9 ± 0.15 b | 3.21 ± 0.66 cd | 0.14 ± 0.01 a | 1.11 ± 0.13 a |
| Adj. 1 m | 5.8 ± 0.03 b | 3.52 ± 0.19 d | 0.14 ± 0.01 a | 1.45 ± 0.30 a |
| Adj. 10 m | 5.8 ± 0.06 b | 2.81 ± 0.29 bcd | 0.11 ± 0.02 a | 2.03 ± 1.09 a |
| *p*-value | * | * | NS | NS |

Means within a column followed by the different lowercase letters are significantly different at $p = 0.05$; SOC, soil organic carbon; Nt, total nitrogen; P$_{av}$, available phosphorus; Values are means ± standard deviations of three replicates. Mint = internal part of the termite mound; Mext = the external part of the termite mound; Adj. 0.5 m, Adj. 1 m, Adj. 10 m represents adjacent soil at the distances of 0.5, 1 and 10 m from the termite mound respectively. * represents significant at $p < 0.05$; NS = not significant at $p < 0.05$.

On Nitisols, termite mounds were relatively depleted in $P_{av}$ and $P_{tot}$ as compared to all adjacent soils (Table 6). A similar trend was observed for acid ammonium oxalate extractable phosphorous ($P_{ox}$), which accounted for 10–20% of soil P. There was no spatial variability with sampling distance from termite mounds. $NH_4$-oxalate extractable iron ($Fe_{ox}$) was significantly lower in the termite mound. The soil P saturation index (PSI %) values were calculated as described by [39], PSI = $((P \div (Al + Fe)) \times 100)$ with P, Fe, and Al concentrations expressed as mmol $kg^{-1}$ to estimate the degree to which P soil sorption sites are filled. Result showed very low P-saturation indices which, in line with $P_{tot}$ and $P_{av}$, were significantly lower in termite mound materials. Unlike on Nitisols, the $P_{av}$ in termite mound (6.5 and 6.7 mg $kg^{-1}$ in inner and external mounds, respectively) was higher than that of the reference soil (5.3 mg $kg^{-1}$) and lower than in adjacent soils at radius of 0.5 and 1 m (11.33 and 9.33 mg $kg^{-1}$, respectively) on Vertisols. Hence, like SOM content, areas within a radius of 1 m from the mounds were relatively hot spots in $P_{av}$ with higher values than reference soil and termite mound materials.

**Table 6.** Phosphorous (activity ($P_{av}$), intensity ($P_{ox}$) and capacity ($P_{tot}$)) and index of phosphate saturation (PSI).

| Site | $Al_{ox}$ (g $kg^{-1}$) | $Fe_{ox}$ (g $kg^{-1}$) | $P_{tot}$ (g $kg^{-1}$) | $P_{ox}$ (g $kg^{-1}$) | $P_{av}$ (mg $kg^{-1}$) | PSI (%) |
|---|---|---|---|---|---|---|
| | | | Limmu kosa, Nitisols | | | |
| Mint | 3.63 ± 0.23 a | 7.11 ± 0.10 a | 0.47 ± 0.03 a | 0.05 ± 0.01 a | 0.00 ± 0.00 a | 0.61 ± 0.1 a |
| Mext | 3.44 ± 0.14 a | 6.84 ± 0.29 a | 0.48 ± 0.00 a | 0.06 ± 0.01 a | 0.19 ± 0.19 a | 0.77 ± 0.1 a |
| Adj.0.5 | 3.15 ± 0.21 a | 11.87 ± 0.97 bc | 0.68 ± 0.02 bc | 0.14 ± 0.01 b | 1.11 ± 0.13 a | 1.39 ± 0.1 b |
| Adj.1 m | 3.38 ± 0.29 a | 12.86 ± 0.80 c | 0.69 ± 0.04 c | 0.15 ± 0.02 b | 1.45 ± 0.30 a | 1.40 ± 0.2 b |
| Adj. 10 | 3.59 ± 0.40 a | 11.92 ± 0.82 bc | 0.62 ± 0.01 bc | 0.15 ± 0.01 b | 2.03 ± 1.09 a | 1.38 ± 0.2 b |
| *p*-value | NS | * | * | NS | NS | * |

Means within a column followed by the different lowercase letters are significantly different at $p = 0.05$; Values are means ± standard deviations of three replicates. * Significant at $p < 0.05$; NS = not significant at $p < 0.05$.

### 3.4. Exchangeable Base Cations and Cation Exchange Capacity

Although slightly lower in the termite mounds in the Nitisols, higher exchangeable cations were measured in termite mounds than adjacent soils at 0.5, 1 and 10 m away in the Vertisols (Table 7), though these trends were not always significant. There also appeared to be a gradual lowering of Ca, Mg and K with distance from the mound in case of the Vertisol sites at Omo Nadda, with eventually 33%, 42% and 45% lowers Ca, Mg and K soil levels at 10 m away from termite mound. In contrast, on Nitisols exchangeable Ca and Mg in termite mounds were depleted by 53%, 50% respectively compared to surrounding soil. K and Na did not differ significantly between mound material and surrounding soil. Mean values of cation exchange capacity (CEC) followed the same patterns as exchangeable base cations on both soil types (Table 7) with respectively higher and lower CEC than surrounding soil in case of Vertisols and Nitisols, respectively.

**Table 7.** Mean values of exchangeable cations and CEC (cmol$_c$ $kg^{-1}$) of termite mound materials and adjacent soils.

| Site | Ca | Mg | K | Na | CEC |
|---|---|---|---|---|---|
| | | | Omo Nadda, Vertisols | | |
| Mint | 24.49 ± 1.25 b | 4.83 ± 0.37 a | 1.24 ± 0.17 a | 0.67 ± 0.16 b | 30.06 ± 0.60 b |
| Mext | 24.44 ± 0.51 b | 4.86 ± 0.51 a | 1.04 ± 0.12 a | 0.66 ± 0.10 b | 29.01 ± 1.01 ab |
| Adj.0.5 m | 22.74 ± 1.1 ab | 4.87 ± 0.34 a | 1.01 ± 0.03 a | 0.28 ± 0.0 ab | 28.56 ± 0.43 ab |
| Adj. 1 m | 20.87 ± 1.8 ab | 4.59 ± 0.13 a | 0.89 ± 0.04 a | 0.22 ± 0.03 a | 27.69 ± 1.60 ab |
| Adj. 10 m | 18.35 ± 1.04 a | 4.03 ± 0.05 a | 0.87 ± 0.06 a | 0.46 ± 0.1 ab | 24.38 ± 1.65 a |
| *p*-value | * | NS | NS | * | * |

**Table 7.** *Cont.*

| Site | Ca | Mg | K | Na | CEC |
|---|---|---|---|---|---|
| | | | Limmu kosa, Nitisols | | |
| Mint | 4.30 ± 0.71 a | 1.63 ± 0.06 a | 1.02 ± 0.17 a | 0.09 ± 0.02 a | 18.60 ± 0.63 a |
| Mext | 4.22 ± 0.15 a | 1.97 ± 0.11 ab | 0.82 ± 0.27 a | 0.07 ± 0.01 a | 19.41 ± 0.24 ab |
| Adj.0.5 m | 6.55 ± 0.13 b | 2.52 ± 0.12 cd | 0.76 ± 0.07 a | 0.09 ± 0.02 a | 21.46 ± 0.25 bc |
| Adj. 1 m | 6.81 ± 0.40 b | 2.52 ± 0.18 cd | 0.68 ± 0.06 a | 0.07 ± 0.01 a | 21.82 ± 0.78 c |
| Adj. 10 m | 6.33 ± 0.27 ab | 2.28 ± 0.1 bcd | 0.69 ± 0.18 a | 0.08 ± 0.01 a | 21.46 ± 0.10 bc |
| *p*-value | * | * | NS | NS | * |

Means within a column followed by the different lowercase letters are significantly different at *p* = 0.05; Values are means ± standard deviations of three replicates. Mint = internal part of the termite mound; Mext = the external part of the termite mound; Adj. 0.5 m, Adj. 1 m, Adj. 10 m represents adjacent soil at the distances of 0.5, 1 and 10 m from the termite mound respectively. * represents significant at *p* < 0.05; NS = not significant at *p* < 0.05.

## 4. Discussion

### 4.1. Selective Enrichment of Mineral Matter in Mounds

The significantly higher clay content in termite mounds is in agreement with the majority of previous research work [40–42]. Literature justifications were preferential selection of clay as cementing materials [21,22] and transport from deep clay rich argic horizon [33,43]. However, it is not clearly explained just why termite transport more subsoil material, including whether it is purposively to find more clay rich material or in search of moist subsoil for mound building. The enrichment of clay in mound material on the studied Vertisols, would favor the hypothesis that termites preferentially select clay because Vertisols are well mixed due to churning or homogenization processes [44] and have no argic horizon. On Nitisols, the very high, clay content in termite mounds are most likely explained by the combination of two processes: (1) transport from clay rich argic subsoil horizon material, indicated by the low OC content of termite mounds, and (2) preferential selection as the only possible explanation for the about 80% clay in termite mounds, not encountered even in the argic horizon of Nitisols. The observed clay enrichment in termite mounds on Vertisols (more that 50% clay in control soils) was in contradiction with former studies [26,45] in which it was concluded that above a certain soil clay content no further enrichment would happen. We postulate that, depending on the combination of soil clay content, mineralogy and in particular climate determine requirement of clay to ensure mound stability and control of internal temperature and humidity. Rainfall is very high at both sites and this likely prompts for the here observed unusually strong clay enrichment to levels (60–80% clay) that strongly enhance particle cohesion and resist dispersive forces of raindrop impact. It would appear logical that there is a larger need for termite mound clay enrichment on Nitisols (100 to 167% clay increase) than on Vertisols (15 to 21% clay increase) due to the much lower surface area and likely cementing action of the Nitisol's kaolinite-dominated clay vs. the Vertisol's smectite-dominated clay.

The selection for clay was not accompanied with a higher content of pedogenic oxides of mound materials versus surrounding soil. This may at first seem surprising because Fe- and Al-oxides are the dominant binding agents in oxide rich soils [46]. On the other hand, the aggregating effect of oxides is mainly at the micro aggregate level [47] and may not be relevant to stabilize mound macrostructures. Also, termites might be incapable of selectively extracting and transporting small and mineral bound Fe-or Al-oxides [46].

### 4.2. Aggregate Stability

In the present study, the lower aggregate stability of termite mound materials on both soil types is against their prominently higher clay content, but in line with the lower levels of pedogenic oxides. Earlier studies reported contradicting finding against the theory that soil organisms increase the stability of soil aggregates [47,48]. Much relates probably to the scale of investigated aggregates as soil-feeding termites only form micro aggregates by passing soil material through their intestinal system and depositing it as fecal pellets or by

mixing the soil with saliva using their mandibles [49,50]. Despite the here and elsewhere found lower (macro-) aggregate stability, termite mounds are stable for decades even after colony death, and local farmers prefer termite mound materials for construction in rural areas. Low penetration of water into termite mounds, higher hydro-repellency of termite mound aggregates [28] and macro scale resistance against mechanical breakdown [22] reconcile this with nevertheless lower aggregate stability. The high difference in aggregate stability observed between termite mound and adjacent soils under fast wetting than slow wetting is reasonably justified by the higher clay content of termite mounds leading to higher volume of entrapped air and differential swelling in fast wetting especially on Vertisols. The larger difference in aggregate stability between slow and fast wetting (SW and FW) in case of Vertisol samples than in counterpart Nitisol samples could be justified by the effect of swelling of 2:1-clays in the former as opposed to non-swelling1:1-clay in the latter soils soil type [51].

### 4.3. Effect on Water Retention

Water retention capacity of the soil is mainly dependent on organic matter and clay content [52]. Our result of higher water retained at FC ($-10$ kPa) and PWP ($-1500$ kPa) in termite mounds than the adjacent soils was a point of discussion as far as clay was higher in termite mounds and higher SOM in adjacent soils. It was reported that the effect of organic carbon on water retention was marginal in medium and fine textured soils [53]. It can also be explained by the magnitude of elevated clay in mounds of termites camped the increased SOM in adjacent soils. Water retention at all matric potential and Plant available water (PAW) was almost two times higher in termite mounds than in adjacent soils on Vertisols, whereas not significant on Nitisols although both soil types have equivalent SOM. Even comparing the two soil types, water retention and PAW was quit high in Vertisols that Nitisols although clay amount is relatively high in Nitisols. Here, it is interesting to note the effect of clay mineralogy was significant on water retention. Despite this fact, the role of clay mineralogy on water retention has received little attention. The inconsistency finding of significantly improved PAW of sandy soils treated with termite mound materials of 200 g kg$^{-1}$ clay content [54] and no improvement with termite mound materials 490 g kg$^{-1}$ clay content [55] also might be due to the difference in clay mineralogy. This presents the need to consider clay mineralogy before suggesting termite mounds to improve PAW retention of sandy soils. On Nitisols, the higher water retention in termite mounds above $-7$ kPa, i.e., at $-1$, $-3$ and $-5$ kPa, relatively higher water retention in adjacent soils is an indication that the effect of SOM and clay is not equal at different matric potential. In agreement to this result, water retention at $-33$ kPa is strongly affected by OM than water retention at $-1500$ kPa [54].

### 4.4. Organic Matter and Total Nitrogen

The low termite mound material OC and N contents for both soil types accords with [22,56,57] but contradict with others who reported, no significant difference [18,25] and opposite trends [26,41]. Explanations for variation in observed differences between mound material and reference soil organic matter content in literature include termite modes of mound constructions and difference in their feeding habits [56,58], salivary or faecal materials that are frequently used as adhesives [59], and selectivity of termites to build mounds with subsoil [43,57]. However, most studies neglected potential control of mounds on the dynamics of C and N in surrounding soil by as a result of land use/land management and saw relative enrichment or depletion only as resultant of termite transport. Our consistent findings of elevated C and N within 1 m radius from the termite mounds compared to both termite mounds and reference soils could not be attributed to mound material erosion, as previously suggested by [29,60]. Alternative possible perhaps simultaneously contributing effects are: (1) locally improved growth of vegetation and biomass C-inputs by a higher soil moisture availability because of run-on from termite

mounds; (2) there may be N fixation by microbes associated with termite mounds, (3) It might be virgin land; hard to cultivate by traditional *maresha*.

For the studied sites it would not be favorable to spread the C and N depleted termite mound material as soil amendment, opposing the recommendations of [6,61]. Besides of their depleted N content, extensive use of termite mounds spread as fertilizers would anyway be limited by the availability of mounds materials and is ecologically likely not sustainable. Alternatively, an accumulation of more C and Nt within but a 1 m radius from the mounds offers but a limited scope for spatially differentiated fertilizer management and arrangement of crops. Nevertheless, in agreement with our findings of elevated C and N near mounds, farmers cultivate crop which require good water and nutrient supply adjacent to termite mounds in Uganda [62].

### 4.5. Effect on Available Phosphorous

Phosphorous is one of the most important macronutrients, the least accessible and hence the most frequently deficient nutrient in many agricultural soils, especially in highly weathered tropical soils. Several literature sources discussed available phosphorous ($P_{av}$) in termite mounds and reference soils and again results were contradictory. The inconsistency could be due to the various sources of P (fertilizer and organic sources) and the complex nature of P with soil minerals and soil reaction. Furthermore, soil particles undergo modifications while pass through the gut of termite because of the alkaline pH and hence affect available P [60]. In our case, the lower $P_{av}$ in termite mounds mostly attributed to their higher clay content. P-absorbing capacity increases with increase of clay content especially in highly weathered tropical soils [44]. Evidently, $P_{av}$ and $P_{ox}$ account only 10% of $P_{tot}$ in termite mounds, whereas, in adjacent soils it ranges from 20% to 24%, showing more pedogenic P fixation in termite mounds than in the adjacent soils. Such low Pav and combined with more pedogenic P fixation in termite mounds constrain the possibility of using termite mound as soil amendments in nutrient deficient and low-input cropping system. Furthermore, the lower ammonium oxalate soil P saturation index (PSI%) in termite mounds is an indication that there will be more fixations of phosphorous if termite mound spreads to the surrounding for soil fertility amendment, compounding the problem of already low Pav on Nitisols. The observation that the $P_{av}$ was a factor 6–10 lowers in the Nitisol than in the Vertisols; although SOM content was comparable further confirm the general finding that clay mineral type has remarkable effect on $P_{av}$. On Vertisols, areas within a radius of 1 m away from termite mounds were hot spots in $P_{av}$ as compared to termite mound and the reference soils is most likely explained by soil pH (pH 7.1 and 6.8 in termite mounds and adjacent, respectively) and the significantly higher SOM within a radius of 1 m from termite mounds. High organic matter is a source of P (both organic P and $P_{av}$) and it may reduce P fixation in soil [63]. Organic acids are known to compete with phosphate for sorption sites on soil mineral surfaces [64] and inorganic P desorbed from clay surfaces by soluble organic acids released from plant residues [63].

### 4.6. Exchangeable Base Cations and CEC

The enrichment of exchangeable base cations and CEC in termite mounds on Vertisols is in agreement with bulks of literature [18,23,65,66]. Various modes of enrichment are mentioned: turnover of some primary minerals from deep saprolite source [60], an "umbrella effect" [67], weathering or modifications of clay mineral by termite activity [22] and nutrients in mounds being not accessible to plants [68]. The lower contents of exchangeable base cations and CEC in termite mounds on Nitisols were unexpected. Because the high rainfall of the study site (2000 mm yr$^{-1}$) is expected to cause leaching on cations in adjacent soil and this would not be in termite mounds due to the above mentioned "umbrella effect". An explanation for this anomaly may be the thick savannah type of vegetation with deep roots has the ability to bring exchangeable cations from deeper soil to the surface and keep in cycling. For the low CEC in termite mounds, the 1:1 clay mineralogy of Nitisols with low charge and low organic matter would be an explanation the exceptionally elevated

exchangeable $K^+$ in the termite mounds can be most likely attributed to weathering or modifications of clay minerals by termite activity [22,69]. During the course of weathering of 2:1 clay, non-exchangeable potassium in the interlayer site of clay particles were released to the soil solution and are held on the exchange complex.

## 5. Conclusions

The study showed termite mounds were significantly enriched in plant nutrients and SOC on Vertisols, whereas not on the Nitisols compared to control soils. Termite mounds constitute only a store of 43.15 and 6.32 kg ha$^{-1}$ of Nt and 639 and 102 kg ha$^{-1}$ of SOC corresponding to 1.25% and 0.27% of the stock of SOC in the upper 15 cm soil layer on Nitisols and Vertisols, respectively. All other nutrients stock in mounds was below 0.05% of in the upper 15 cm topsoil. Thus, under the studied condition, spreading mound materials for soil fertility amendment will not be suggested. Furthermore, on Nitisols, the higher clay in termite mounds will probably complex the already limiting Pav. This study revealed that the need to accurately quantify the amount of nutrient stock in termite mounds per area and nutrient concentration in the termite mounds alone is not be sufficient to suggest use of termite mounds for soil fertility improvement. Interestingly, on both soil types, the immediate vicinity of termite mounds was a fertile hotspot compared to termite mound and control soils although the exact process is not fully understood. This suggests spatial arrangement of crop around termite mounds according to soil fertility gradient. Comprehensive study of nutrient concentration and their accurate quantification and soil spatial variability around termite mounds help the decision whether to level mounds or guide nutrient management strategies on the basis of niche fertility. Interesting topics for future investigation would be growth experiment for different crop with mound materials treatment. It would also be interesting to study the effect mound building termite on soil properties under different soil condition, slope class and land management/use.

**Author Contributions:** Conceptualization, A.T., W.C. and E.V.R.; methodology, A.T. and W.C.; software, A.T.; validation, A.T. and S.S.; formal analysis, A.T. and A.N.; investigation, A.T.; resources, B.D.; data curation, A.T. and S.S.; writing—original draft preparation, A.T. and S.S.; writing—review and editing, W.C., S.S., E.V.R. and A.N.; visualization, A.T.; supervision, W.C., S.S. and E.V.R. All authors have read and agreed to the published version of the manuscript.

**Funding:** This paper is drafted from MSc thesis funded by the VLIR-UOS MSc scholarship, Ghent University, Belgium.

**Institutional Review Board Statement:** Not applicable.

**Data Availability Statement:** Research data in this paper can be requested from the corresponding author. But are not publicly available due to privacy restrictions.

**Acknowledgments:** Special thanks go to Flemish Inter-University Council (VLIR) for all the financial support throughout the study programme. Without you Amsalu Tilahun would not have had an opportunity to stay in Belgium, Gent University. Amsalu Tilahun also would like to express his grateful to Jimma University, College of Agriculture and Veterinary Medicine who granted him study leave. Thanks also go to Ghent University, Department of Soil Management soil laboratory staff specially Maarten Volckaert and Luc Deboosere for their guidance during the soil laboratory work. We thank Amsalu Abera for his technical advice during research site selection and transport support. We also thank MitikuTezera for his field assistance during data collection and Melkamu Dumesa for language edition.

**Conflicts of Interest:** The authors declare that they have no conflict of interest.

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
