# Peer review of "The Potential of Termite Mound Spreading for Soil Fertility Management under Low Input Subsistence Agriculture"

_agriculture, doi:10.3390/agriculture11101002_

Round 1

Reviewer 1 Report

Dear authors,

Thank you for the nice submission. It was easy to read, and clearly presented. I have only 1 major point concerning the content, all other comments are minor, and concern layout/spelling/minor mistakes

Major point of concern: determination of termite species

While the termite mounds are well described (height, volume), the termites itself are barely introduced. For example, as the authors know, there are many different type of termites. Are all termite mounds from the same species? Were they determined? Were all mounds active? And (important for this research), what kind of feeders are they (soil-feeders, soil-wood feeder, fungi-feeders?).

The best would be if the authors determined the termite species, and could name the specie(s), and describe its feeding habits. In case the authors did not do this: I can imagine that the fieldwork part of this research is over, and that this might be difficult to retrieve this information afterwards. In that case, this part can be complemented by a small literature review. Which termite species are common in this area? Add some references from local entomology studies if applicable/possible.

To sum up: the authors should minimally inform if all mounds were active/abandonned, and should state if the termite species was the same everywhere. Then, if possible, state the termite specie(s) and their feeding habits or (if not possible) elaborate a little bit on the types of termites present in this ecosystem by a small literature review of local literature. Also, in the discussion, if the termites were different at the different field sites, discuss this briefly. Could the found differences be explained by this?

Minor point to add to introduction

The authors name the different soils (vitisols and nitisols). Could the authors add 1 or 2 sentences quickly explaning what these terms mean? What is the main difference between the soils? This will help the readers who are not so much into soil science.

Minor comments:

line 77&80: twice use of 'on the other hand'--> that reads weird

line 97: pick rainfall, should be peak rainfall?

line 203-204: Is this sentence correct? seems that a verb is missing

fig 2, in legend, change writing of Mint and Mout (maybe write 'M_int, and M_out'). Also, there is a white space in the legend (right figure)

line 374, white space missing

reference at line 403, bracket wrong
line 5 & 450 & 454, a white space too many

References

check your references. I saw Six et al 2004 in the text, but not in the list. 
Maybe there are more of these mistakes? Check thorougly.

Acknowledgements

Does this journal do not have acknowledgements? I assume that this study had quite some fieldworkers involved, which are not co-author, which can be named here. And any funding agencies which need to be thanked?

Author Response

Dear reviewer, 

We are happy with the nice comments and suggestions on our manuscript. Thank you for that. Hopefully we have addressed the comments and suggestions forwarded and are attached herewith

Best regards,

Reviewer 2 Report

Comments and suggestions for Authors

”The potential of termite mound spreading for soil fertility management under low input subsistence agriculture”

The manuscript present on physical and chemical properties of the mounds in relations to the surrounding soils and accurately quantified the nutrient stored in termite mounds to evaluate the potential of termite mounds as an alternative means for soil management in Jimma area, South west Ethiopia. Subject is interesting and fall within the scopoe of the journal. The experimental dataset undoubtedly are useful experiments and constitutes some scientific values.

Metodology of the article is very good, the methods were chosen well. The results are clearly presented and supported by arguments.

The following points may be addressed by the Authors to enhance the usefulness of the paper.

Remarks

  1. Line 29 and 30 - you should increase the number of keywords.
  2. A map should be used to present the location of the research.
  3. Line 215 - maybe better 1.67 and 1.22 times.
  4. Table 5 - At pH values should not count average and thus SD.

Line 4 – it should be: Abebe Nigussie

Line 56 – to be complet Kaschuk et al. 2006

Line 149 Kemper and Rosenau (1986) - It is not in the list of references.

Line 271 Zhang et al. (2005) - It is not in the list of references.

Line 313 (WRB, 2006) - It is not in the list of references.

Line 332, 334 Six et al. 2004; Oades et al. 1989 - It is not in the list of references.

Line 345 Jungerius et al. 1999; Bignell and Holt 2002 - It is not in the list of references.

Author Response

(The authors gave the same response as above.)
